# Return to Physical Activity in Individuals with Surgical Stomas: A Scoping Review

**DOI:** 10.3390/sports12100273

**Published:** 2024-10-10

**Authors:** Andrea-Victoria Mena-Jiménez, Claudio-Alberto Rodríguez-Suárez, Héctor González-de la Torre

**Affiliations:** 1Insular Maternal and Child University Hospital Complex, Canary Health Service, 35016 Las Palmas de Gran Canaria, Spain; andrea.mena102@alu.ulpgc.es; 2Nursing Department, Faculty of Healthcare Science, University of Las Palmas de Gran Canaria (ULPGC), 35016 Las Palmas de Gran Canaria, Spain; hector.gonzalez@ulpgc.es

**Keywords:** exercise, return to sport, surgical stomas, ostomy, quality of life, review

## Abstract

In surgically treated individuals with surgical stomas, the return to physical activity is an indicator of quality of life that reflects their well-being. With the aim of synthesizing the available evidence regarding the return to physical activity in individuals with surgical stomas, a scoping review was developed following the methodological approach of the Joanna Briggs Institute and the Preferred Reporting Items for Systematic reviews and Meta-Analyses for Scoping Reviews criteria. Searches were conducted in Medline (PubMed), Scopus, Web of Science, Cinahl, and Lilacs, as well as the meta-search engines TripDatabase and Epistemonikos, using MeSH terms. Included studies were written in Spanish, English, Portuguese, and German, without any limitation on the year of publication. A total of *n* = 15 studies was included (*n* = 2 qualitative; *n* = 2 case reports; *n* = 1 case series; *n* = 1 cohort; *n* = 8 cross-sectional; and *n* = 1 randomized clinical trial), which showed variability in the quality of the designs. The qualitative studies explored themes such as motivation, beliefs about physical activity, and other lifestyle factors. The case reports described physiological, psychological, and functional implications of returning to physical activity for specific individuals after ostomy surgery. Quantitative studies evaluated the effects of different types of physical activity on quality of life and tolerance to physical activity in these individuals, employing various measurement instruments. In conclusion, the evidence on returning to sports and physical activity after stoma surgery is limited and varied. While studies highlight the importance of social support and self-confidence, they generally lack rigor and primarily focus on adults and oncology patients. There is a need for more research to establish clear guidelines on physical activity type, frequency, and intensity to ensure safe and beneficial outcomes for individuals with stomas.

## 1. Introduction

Cancer is a disease characterized by the uncontrolled growth and spread of abnormal cells within the body [1]. Colorectal cancer, in particular, affects the cells of the colon or rectum. According to the World Health Organization (WHO), colorectal cancer is the fourth most common type of cancer globally, with a surgical intervention rate exceeding 90%, often requiring a temporary or permanent ostomy [2]. It is estimated that colorectal cancer accounts for 50% of intestinal stomas [3]. Furthermore, bladder cancer is the second most common urinary tract malignancy, mostly affecting patients aged 65 years or older [4], and may also require the creation of urinary stomas. However, there are a number of non-oncological pathologies or complications of diseases that require the creation of a stoma in both adults and children [5].

An ostomy is a surgically created opening between a hollow organ and the body’s surface, primarily intended to divert intestinal contents [6]. There are two main types of intestinal stomas: colostomies and ileostomies. Colostomies are classified as permanent or temporary based on the duration or reversibility of the ostomy, and as ascending, transverse, descending, or sigmoid according to the part of the colon used. Ileostomies can also be permanent or temporary and, depending on the surgical technique, may consist of a conventional or Brooke ileostomy, a continent ileostomy, or an ileoanal reservoir (J-pouch or pelvic pouch) [7]. In relation to the urinary stomas, the urinary diversion can be divided into two types according to their urine reservoir manipulation: continent reservoirs and non-continent reservoirs [5].

The creation of a stoma involves economic, physical, psychological, and social challenges for individuals, significantly affecting their quality of life (QoL). Patients with a stoma often experience a lower QoL compared to those without one [3]. QoL has been described as the overall well-being of an individual or community, considering their health and level of happiness rather than focusing exclusively on economic factors. Additionally, it can be understood as the patient’s perception of their physical, psychological, social, and spiritual health [7]. Changes in the QoL of individuals can result in issues such as anxiety, sleep disturbances, and difficulties in interpersonal relationships [8]. Living with an ostomy leads to changes in body image and functionality, impacting personal aspects such as sexuality, as well as occupational and social factors. These include challenges in daily activities, managing stoma-related complications, finding privacy to empty the pouch, and coping with odors from fecal leaks [9,10]. In these individuals, it is crucial to promote and develop their self-care capabilities to reduce such clinical complications and social health issues [3]; among these interventions, actions aimed at physical activity (PA) should be included [10].

PA is essential to enhancing QoL as it integrates physical, mental, and social dimensions to promote and sustain an active, healthy lifestyle. PA is a comprehensive term encompassing various forms of movement, defined as any bodily movement produced by skeletal muscles that results in energy expenditure above basal levels. Exercise and sport are distinct subcomponents of PA. Exercise typically involves structured, purposeful, and repetitive isometric or isotonic movements aimed at maintaining or improving physical fitness [11]. In contrast, sport generally refers to competitive forms of PA, which are regulated by rules and guidelines. Beyond its physical benefits, PA exerts positive effects on mental health by reducing stress levels and promoting psychological relaxation. Moreover, it provides social benefits, particularly when performed in group or team-based settings [11].

In individuals with stomas, the return to PA may be beneficial for aerobic fitness, cancer-related fatigue, and health-related QoL [12,13], including physical functioning, role function, social functioning, and fatigue [13]. However, clinical outcomes related to the number of complications, incidence of parastomal hernias, or prolonged hospital stays have not yet confirmed these positive effects [3]. In this context, while the return to PA is often used as an indicator to assess orthopedic clinical outcomes in sports medicine, it is frequently interpreted simply as the resumption of PA [14]. Most studies have focused on athletes who have already returned to competitive sports or resumed training. However, the lack of standardization or structured programs during this return affects the accurate evaluation of outcomes [15,16]. Thus, the available literature on returning to sports after surgery is extensive, particularly in the areas of sports traumatology [17,18,19,20] and pediatrics [21,22]. However, the topic of resuming PA in individuals with stomas remains unexplored. Currently no reviews on PA and individuals with stomas have been developed. To address this gap, a scoping review has been conducted to examine the existing scientific literature. The research question guiding this review is: What evidence is available regarding the return to PA in individuals with surgical stomas within the context of post-surgery rehabilitation? The aim was to synthesize the existing evidence regarding the return to PA in individuals with surgical stomas.

## 2. Materials and Methods

### 2.1. Design

A scoping review was conducted according to the Preferred Reporting Items for Systematic Reviews and Meta-Analyses for Scoping Reviews (PRISMA-ScR) criteria [23]. To develop the review approach, the methodological proposal of the Joanna Briggs Institute (JBI) was followed. The research question followed the structure comprised of Population (P): Individuals with surgical stomas; Intervention (I): Return to PA; and Outcome (O): Post-surgery rehabilitation (in terms of physical and mental health). The review protocol was registered on 1 November 2023, prior to starting the review process, in the Open Science Framework (OSF), and can be accessed at the following link: https://osf.io/7yv6d/ (accessed on 1 July 2024).

### 2.2. Information Sources

As a first step, the scientific literature was consulted to determine the existence or not of systematic reviews addressing the topic and a search was conducted in PROSPERO to find registered protocols corresponding to surveys that answered the same research question. After this initial query, searches were conducted in the following databases: Medline (through PubMed), Scopus (through Scopus-Elsevier), SCI Expanded (trough Web of Science [WoS]), Cinahl (through EbscoHOST), and Lilacs (through Virtual Health Library). Searches were also conducted through the meta-search engines TripDatabase and Epistemonikos; records retrieved from these latter sources were considered as free searches.

### 2.3. Search Strategies

The searches were conducted between 2023 October and 2024 March using the MeSH terms: “return to sport”, “exercise”, “sports”, “surgical stomas”, “ostomy”, “colostomy”, “ileostomy”, “enterostomy”, “cecostomy”, “duodenostomy” and “jejunostomy”, or entry terms combined with the Boolean operators AND and OR. The searches were performed by one of the researchers (A.-V.M.-J.) and verified by a second (C.-A.R.-S.) using PRISMA-S for search extension [24]. The final search strategy established was adapted to each of the databases (Appendix A).

### 2.4. Eligibility Criteria

Inclusion criteria were as follows: According to the population: Adults and children with surgical stomas who have addressed the return to PA. According to the study design: Publications with any methodology (qualitative or quantitative) and design (experimental, observational, or qualitative) were included, as well as case studies and case series. These publications were available in Spanish, English, Portuguese, and German, with no limits on the year of publication. The broadness of these criteria is due to the fact that the review is exploratory with the purpose of finding out the state of the science on this subject. Exclusion criteria were as follows: According to the study design: Publications that did not correspond to research studies, such as opinion articles, editorials, and letters to the editor. Grey literature and research study protocols were also excluded.

### 2.5. Screening Process

After performing the searches, duplicate records were eliminated and screened by title and abstract. The full-text documents of the selected records were then retrieved to assess their eligibility according to inclusion and exclusion criteria. The JBI critical appraisal tools appropriate to each design were used. Screening was performed independently by two reviewers (A.-V.M.-J. and C.-A.R.-S.) and, in case of discrepancies, a third reviewer decided (H.G.-d.l.T.). As a scoping review, the critical appraisal process was not used to eliminate low-quality studies, but to identify and establish the quality of the included studies [25]. A pilot phase was conducted with a sample of records to assess the adequacy of the process. One reviewer applied the specific JBI tool to each design, while a second reviewer independently verified the accuracy of the extracted data. The item-by-item quality assessment from JBI is available as Appendix A.

### 2.6. Data Extraction

The primary outcome variable of the study was the return to PA in individuals with a stoma. However, data on the return to PA were also extracted when reported as a secondary outcome. As a secondary outcome variable, information on QoL and body composition was extracted when reported. Bibliometric (authors, year, and study design) and sociodemographic (country, age of participants, type of disease, setting, and type of stoma) variables from the studies included were extracted, as well as the descriptive (data collection, sample and patients characteristics, themes and sub-themes, measuring instruments, and PA characteristics), and statistical data (mean, standard deviation, percentage, odds ratio, confidence intervals, effect sizes, or other additional statistical results). Data were exported to an Excel^®^ (version: 16.17) file, with methodological support provided by the Mendeley^®^ reference manager software (version: 2.121.0). Data extraction was carried out independently and blinded by two researchers (A.-V.M.-J. and C.-A.R.-S.), and a third reviewer settled any and all discrepancies (H.G.-d.l.T.). When necessary, authors were contacted to complete the information that was not retrieved. A pilot phase of the data extraction process was conducted using a sample of studies through the development of a data extraction table in Excel^®^, which incorporated the key information outlined above. Additionally, a free-text field was included to capture any unexpected or Appendix A. Initially, one reviewer performed the data extraction, while a second reviewer independently verified the accuracy of the extracted data for each study design group.

## 3. Results

A total of *n* = 1802 records was identified. After removing duplicates (*n* = 921), the number of records that were screened by title and abstract was *n* = 881, excluding *n* = 844 records; the remaining *n* = 37 were retrieved for quality assessment and full-text screening, with *n* = 15 studies included in the review, as shown in Figure 1.

Among the studies excluded, *n* = 22 did not meet the inclusion criteria (Appendix A).

Table 1 shows the general characteristics, aims, conclusions, and methodological quality after applying the JBI critical appraisal tools.

The included studies comprised qualitative (*n* = 2), cohort (*n* = 1), cross-sectional (*n* = 8), case studies (*n* = 2), case series (*n* = 1), and randomized clinical trial (RCT) (*n* = 1) designs, showing variability in the quality of the studies. Most of them were conducted in the United Kingdom (UK) (*n* = 7), United States of America (USA) (*n* = 4), and Germany (*n* = 3), with *n* = 1 study from Canada. Only the study by Goodman et al. [27] included adolescent patients, while none involved children. Regarding the type of disease, all studies included cancer survivors, except Park et al. [26], Goodman et al. [27], and Russell [33,34], which also involved non-cancer patients. Finally, Lowe, Alsaleh, and Blake [31] did not specify the underlying pathology that led to the stoma in their study.

Two studies did not clearly specify the qualitative approach in their methodology and utilized different data collection techniques. The themes explored included motivation, beliefs about PA, and other lifestyle factors. The emergent themes and sub-themes are shown in Table 2.

On the other hand, two case report studies [36,37] and one case series report [40] describing the evolution process of the ability of these individuals to return to PA after the ostomy were included, in which physiological, psychological, and functional implications were addressed, as shown in Table 3.

Regarding the quantitative studies, only Mo et al. [29] reported 150 min of moderate exercise or 75 min of intense exercise. Lowe, Alsaleh, and Blake [31] mentioned walking, while Kindred et al. [32] referred to strength exercises. The remaining studies did not specify the type of exercise. PA was evaluated using different validated instruments, such as the International Physical Activity Questionnaire (IPAQ) short form [26,31], the Excess Benefits and Barriers Scale (EBBS) [31], the Body Esteem Scale (BES) [32], and the Godin Leisure-Time Exercise Questionnaire (GLTEQ) [39]. Additionally, modified instruments were used, including Self-Efficacy to Perform Self-Management Behaviors [29] and Self-Efficacy for Exercise (SEE) [31], as well as ad hoc instruments [33,34], as shown in Table 4.

## 4. Discussion

Given the limited number of available studies, it is important to note that some exhibit low methodological rigor, such as the cross-sectional study by Russell [33,34], which received very low scores (25%). Other studies, such as those by Park et al. [26], Goodman et al. [27], Krogsgaard et al. [28], and Courneya et al. [39] are cross-sectional studies with secondary analyses of primary results, potentially representing redundant publications or a “salami slicing” strategy in scientific publishing [41]. Furthermore, the cohort study by Courneya et al. [39] was conducted over 20 years ago, and therefore does not account for recent advancements in this clinical field. Additionally, Wiskemann et al. [36] and Sica [37] presented case reports, while Isaacs [40] conducted a case series study. Isaacs’ case series, published in 1984, also lacks more recent updates to validate its findings or incorporate newer, more sensitive analytical techniques. Of the included studies, only one was an experimental study [32] and another an analytical observational study [39], both of varying quality. The remaining studies include qualitative designs [30,38], which provide lower levels of evidence. Therefore, further research is needed to develop specific recommendations that promote healthy lifestyles incorporating PA, and to better understand its impact on the QoL for individuals with stomas.

In this context, the number of studies addressing the return to PA in patients with surgical stomas is scarce. In contrast, numerous publications have focused on aspects related to sexuality [42]. It has been noted that the experiences of individuals with stomas significantly affect their self-perception and behavior due to alterations in body image, lifestyle restrictions, and the need to overcome these challenges [43]. Individuals with stomas must learn to recognize and adapt to these changes, making behavioral adjustments to manage the associated restrictions. This process often involves deciding whether to disclose or conceal their stomas, based on concerns about social acceptance or potential rejection. This process involves optimizing internal resources and proactively seeking external support [13,35]. Thus, Mulchandani et al. [44] showed that PA interventions in the workplace have positive effects on metabolic and cardiovascular health, significantly reducing body weight (16 studies; *mean difference* = −2.61 kg, 95% *CI*: −3.89 to −1.33), body mass index (BMI) (19 studies; *mean difference* = −0.42 kg/m^2^, 95% *CI*: −0.69 to −0.15), and waist circumference (13 studies; *mean difference* = −1.92 cm, 95% *CI*: −3.25 to −0.60). Similarly, Reed et al. [45] found significant reductions in body weight (7 studies; *mean difference* = −0.83 kg; 95% *CI*: −1.64 to −0.02), BMI (6 studies; *mean difference* = −0.35 kg/m^2^; 95% *CI*: −0.62 to −0.07), low-density lipoprotein (4 studies; *mean difference* = −0.11 mmol/L; 95% *CI*: −0.17 to −0.04), and blood glucose (2 studies; *mean difference* = −0.18 mmol/L; 95% *CI*: −0.29 to −0.07). However, Mulchandani et al. [44] noted reductions in blood pressure, lipids, and blood glucose were not statistically significant.

From the perspective of professionals who care for individuals with stomas, describing and interpreting their personal awareness and associated behavioral choices is essential for offering practical, informational, and emotional support during the adaptation process [3,43]. This understanding helps professionals tailor their care to better address the unique challenges these individuals face.

Regarding the qualitative findings, the studies reviewed explored the experiences and perceptions of individuals with stomas. While these results are transferable to clinical practice, they are not generalizable due to the subjective nature of the data collected. Although these studies offer valuable insights into individual cases, their broader applicability is limited. Moreover, the lack of a clearly defined methodological framework in many of these studies’ designs reduces the credibility of the findings. The evidence suggests that PA may influence QoL in stoma patients, though outcomes vary based on factors like the type of stoma. Thus, Saunders and Brunet [30] and Anderson et al. [38] identify several categories related to motivations, beliefs, barriers, social support, and perceived needs concerning PA and lifestyle in individuals with stomas, which impact the reduction in long-term complications. Both studies highlight the importance of understanding the motivations and perceived barriers related to engaging in PA and maintaining a healthy lifestyle, as well as the need for practical guidance and support to help these individuals adopt healthier behaviors. Despite their shared focus, the studies differ in approach: Saunders and Brunet [30] emphasize individual’s immediate experiences, while Anderson et al. [38] explore long-term expectations.

Secondly, the case studies also reveal the individual impact of PA and training programs on QoL, physical function, and psychosocial adaptation in individuals with stomas. Specifically, the cases presented by Wiskemann et al. [36] and Sica [37] highlight the return to PA in individuals with stomas following oncological treatments. In contrast, Isaacs [40] examines the effects of biochemical blood concentrations in athletes with stomas after high-intensity exercise. Collectively, these case studies underscore the importance of PA and supervised training programs in enhancing the adaptation and well-being of individuals with stomas. However, the findings are based on individual experiences and are not generalizable.

Finally, quantitative studies offer insights into the relationship between PA, QoL, and other factors in patients with stomas. According to Mo et al. [29] and Krouse et al. [35], active individuals report a better QoL compared to those who are inactive. Goodman et al. [27] categorize QoL into four profiles: good QoL, some problems with QoL, low QoL, and economic concerns. Lowe et al. [31], Kindred et al. [32], and Russell [33,34] do not provide specific information on QoL. Parastomal hernia, a common late complication after stoma formation, significantly decreases QoL and negatively affects body image and physical function [46,47]. Park et al. [26] found significant differences in QoL related to the presence of a hernia (*U* = 11.99; *p* = 0.004), with lower QoL observed in patients with parastomal hernias, as also noted by Krogsgaard et al. [28]. According to Haas et al. [48], parastomal hernia is a common complication following ostomy surgery, with an incidence rate of up to 78%. Most individuals develop a parastomal hernia within two years after surgery. Those with a parastomal hernia may experience pouch leakages, which can lead to skin damage and difficulties in maintaining a secure seal for the ostomy pouching system. They might also suffer from bowel obstructions and chronic pain in the abdomen, back, or hips. In rare cases, bowel strangulation and ischemia can occur. In addition to these complications, living with a parastomal hernia can significantly impact a person’s QoL, contributing to stress related to body image, fatigue, and the physical burden of the hernia.

Evidence indicates that higher BMI, older age, female sex, larger stoma aperture, trans-peritoneal stoma creation, and greater waist circumference are risk factors for parastomal hernias [47]. Engaging in PA under professional supervision has been shown to provide beneficial physical effects for preventing parastomal hernias, as well as psychological benefits for patient recovery [49,50]. Thus, considering PA as an integral part of the care and treatment for stoma patients is crucial to minimize the risk of complications and enhance QoL. Additionally, PA may affect the patient’s ability to adapt to and tolerate stoma [46,50], highlighting the importance of professional support and counseling [51,52,53].

The strengths of this study include the absence of prior reviews on this specific topic, enabling the identification of knowledge gaps related to the return to PA in stoma patients. Additionally, a critical appraisal was conducted based on the inclusion criteria, allowing for an evaluation of the quality of the available studies. However, the methodological limitations identified through the appraisal process pose challenges to the internal validity and reliability of the results. Despite these limitations, several studies were rated as high quality. In terms of review limitations, it is important to highlight that heterogeneity in study designs is a common limitation of scoping reviews, particularly on less-explored topics. Additionally, there is a potential for contextual bias since all the included studies were conducted in Europe, Canada, and the USA, where lifestyle and social norms related to PA may differ. Due to constraints on the length of the study, the critical appraisal process was not fully incorporated into the results section. Given the heterogeneity in study designs and evidence levels, the findings should be interpreted cautiously, considering the inherent limitations of the available data. Moving forward, more rigorous research designs, such as RCTs or longitudinal observational studies, are needed to enhance our understanding of the relationship between surgical stomas and PA. Moreover, since most studies have focused on adults and oncology patients, future research should be extended to include children and non-oncology patients to achieve a more comprehensive understanding of this phenomenon.

## 5. Conclusions

The available evidence on the return to PA in individuals with surgical stomas within the context of post-surgery rehabilitation is limited and heterogeneous. While studies highlight concerns related to physical and psychological complications, as well as the importance of social support, counseling, and self-confidence in facilitating PA, there is a notable lack of rigorous, high-quality research. Most of the existing studies focus on adults and oncology patients, with few addressing children or non-oncology populations. Additionally, few studies provide robust evidence on the specific benefits and challenges of PA for QoL in this population. It is necessary to more clearly define the type, frequency, and intensity of exercise to ensure that the return to PA is both beneficial and free from stoma-related complications.

Further research, including RCTs and longitudinal studies, is needed to better understand the role of PA in improving the overall well-being of individuals with stomas. Future studies should focus on developing valid and reliable measurement tools and expanding research beyond the physical dimension to include psychological, emotional, and social factors. This will enable more comprehensive recommendations for PA in stoma patients, improving their QoL post-surgery.

## Figures and Tables

**Figure 1 sports-12-00273-f001:**
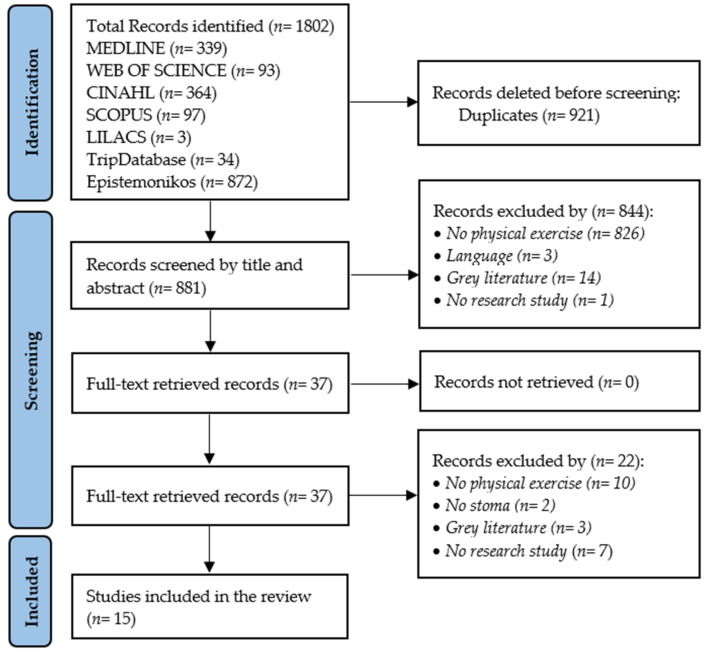
Flowchart of the studies included in the review.

**Table 1 sports-12-00273-t001:** General characteristics of included studies.

Author (Year); Design; Country (Age of Participants) Type of Disease	Aim	Conclusions	JBI ^1^ Quality Items (%)
Park et al. (2023) Cross-sectional; USA (Adults) oncology/non-oncology [26]	To determine rates and risk factors for parastomal hernias in patients with permanent ostomies.	Parastomal hernia rates remain high in current surgical practice. There is an association between PA ^2^ and the presence of parastomal hernia, with a higher rate among those who exercise less.	5/8 (62.5)
Goodman et al. (2022) Cross-sectional; UK (Adolescents/Adults) Oncology/Non-oncology [27]	To identify subgroups in the QoL ^3^ of people with stomas. To assess whether belonging to these groups is associated with demographic, clinical, and PA characteristics.	Some latent profiles associated with demographic and clinical variables were identified, but additional variables should be identified in the future to provide the basis for targeting and tailoring future interventions to specific subgroups of people with a stoma.	5/8 (62.5)
Krogsgaard et al. (2022) Cross-sectional; Germany (Adults) Oncology [28]	To examine the level of PA and explore factors influencing PA in cancer survivors with a stoma.	Half of patients met or exceeded the guideline recommendations. Of the patients who did not meet the recommendations, some could meet them by slightly increasing moderate or vigorous activity.	8/8 (100)
Mo et al. (2021) Cross-sectional; USA (Adults) Oncology [29]	To describe lifestyle behaviors and their relationships with health-related QoL in cancer survivors with ostomies.	Improved QoL is associated with adherence to PA guidelines among cancer survivors with ostomies. Results have clinical relevance for ostomy self-care and establishing lifestyle recommendations.	8/8 (100)
Saunders and Brunet (2019) Qualitative; Germany (Adults) Oncology [30]	To explore the experiences of rectal cancer survivors with a stoma and the impact on their engagement in PA.	The stories provided experiences related to cancer and the stoma, highlighting reasons for and barriers to PA. The results helped to identify useful strategies for those learning to be physically active with a stoma.	7/9 (77.7)
Lowe, Alsaleh, and Blake (2019) Cross-sectional; UK (Adults) Not specified [31]	To assess PA levels in adults with a stoma. To investigate the relationship between activity levels, exercise self-efficacy, perceived benefits and barriers to exercise, depression, body image, and stoma-related QoL.	Most participants were physically inactive. Interventions that reduce barriers to exercise and support self-efficacy in people with stoma may help them increase their PA levels, as well as reduce the risk of chronic diseases associated with sedentary lifestyles.	8/8 (100)
Kindred et al. (2019) RCT; USA (Adults) Oncology [32]	To examine the relationship between changes in fitness and body fat with changes in body esteem among colorectal cancer survivors after testing the effects of a PA intervention.	Improving physical fitness and body composition may enhance self-esteem among these cancer survivors; however, there are differences according to gender and stage of disease.	11/13 (84.6)
Russell (2017) Cross-sectional; UK (Adults) Oncology/Non-oncology [33]	To investigate the physical health and well-being of people living with stomas in the UK.	There are gaps in care regarding advice on PA, abdominal exercises, and prevention and treatment of stoma hernias. More research and training for patients and healthcare professionals is needed.	2/8 (25)
Russell (2017) Cross-sectional; UK (Adults) Oncology/Non-oncology [34]	To investigate the physical health and well-being of people living with stomas in the UK.	PA levels drop significantly after stoma surgery; the vast majority of people living with a stoma do not meet PA guidelines. Those diagnosed with cancer or parastomal hernia are even less active.	2/8 (25)
Krouse et al. (2017) Cross-sectional; USA (Adults) Oncology [35]	To examine the relationships between PA, health-related QoL, and bowel function in rectal cancer survivors.	Meeting or exceeding PA guidelines was associated with increased QoL. The results suggest that women may benefit from increased PA, while men with ostomies may face challenges that require further study. There is a need to identify PA strategies that improve compliance and benefits in patients.	8/8 (100)
Wiskemann et al. (2016) Case report; Germany (Adult) Oncology [36]	To report on how a firefighter with rectal carcinoma and an ostomy was trained to recover fitness for work.	Colorectal cancer survivors with ostomies may be able to recover fitness for demanding physical tasks, such as firefighting, through an individualized and supervised training program.	8/8 (100)
Sica (2016) Case report; UK (Adult) Oncology [37]	Not reported.	Having a stoma and recovering from major surgery can be a challenging and lengthy process. Finding the right pouching system is an integral part of recovery and can help give subjects the confidence to start resuming previous activities, including sport.	5/8 (62.5)
Anderson et al. (2013) Qualitative; UK (Adults) Oncology [38]	To explore perceived needs for advice on diet, activity, and beliefs about the role of lifestyle in reducing disease recurrence.	Lifestyle changes can lead to perceived blame and stigmatization. Personalized, evidence-based counselling on lifestyle choices appears to be a much-needed part of care planning and should be incorporated into survivorship programs.	7/9 (77.7)
Courneya et al. (1999) Cohort; Canada (Adults) Oncology [39]	To examine the relationship between physical exercise and QoL in patients with colorectal cancer.	Small changes in exercise from pre-diagnosis to post-surgery are positively associated with QoL in patients with colorectal cancer, but experimental research is needed before definitive conclusions can be drawn.	5/11 (45.4)
Isaacs (1984) Case series; UK (Adults) Oncology [40]	To detail fluid and electrolyte balance data in a team of ileostomized marathon runners who regularly run long distances.	Healthy ileostomates after adequate training are successful marathoners, but the prevalence of a slight depletion in sodium level in ileostomates suggests that it may also be advisable for them to take glucose or electrolyte solutions when competing at any ambient temperature or when preparing for a marathon in hot environments.	6/10 (60)

^1^ JBI: Joanna Briggs Institute; ^2^ PA: Physical Activity; ^3^ QoL: Quality of Life.

**Table 2 sports-12-00273-t002:** Qualitative studies included in the review.

Author (Year) Design	*N* (Type of Stoma) Data Collection	Themes and Sub-Themes
Saunders and Brunet (2019) Qualitative (No specified approach) [30]	*N* = 15 (Not specified) Semi-structured interviews	*Themes*: Reasons for physical activity *Sub-themes*: Fun, health benefits (mental and physical), sense of accomplishment, weight control, sense of normalcy, taking time for themselves away from daily responsibilities
		*Themes*: Physical activity discourages *Sub-themes*: Negative side effects of cancer and treatments, uncertainty in unfamiliar environments, physical restrictions, fear of injury, unclear orientation, stoma, shyness in public and private, negative previous experiences
		*Themes*: Implications for practice *Sub-themes*: Social support and support networks, counseling, previous experiences, experimentation, safe environment, skills and confidence
Anderson et al. (2013) Qualitative (No specified approach) [38]	*N* = 40 (Not Specified) 6 focal groups	*Themes*: Beliefs about the role of diet, physical activity and lifestyle in reducing long-term disease risk
		*Themes*: Health maintenance actions
		*Themes*: Interest in receiving guidance on diet, activity and lifestyle to reduce risk and disease progression
		*Themes*: What are the forms, schedules and modes of guidance on nutrition, physical activity, and lifestyle?

**Table 3 sports-12-00273-t003:** Cases studies included in the review.

Author (Year) Design	Type of Stoma Patient Characteristics	Case Results
Wiskemann et al. (2016) Case report [36]	Permanent colostomy. Male, 44 years old. Rectal adenocarcinoma. Firefighter. Performed aerobic exercise (running, swimming, rowing) and resistance (weight training with weights and machines). Since the diagnosis has been inactive. To return to work must pass ergometric tests. Training program, 9 months: 0–2 preparation: 1.3 sessions/week. 2–4 supervised individual training: 2.3 sessions/week. 4–6: group training: 3 sessions/week. 6–9: autonomous training.	*European Organization of Research and Treatment in Cancer Questionnaire about Quality of Life 30 (EORTC QoL 30)*: Score: 66.67. Pain and diarrhea decreased 50%, insomnia and loss of appetite disappeared stool control at 9 months. Subscales showed reduction in fatigue (30% mental fatigue, 64% physical fatigue) during first 4 months. Distress was reduced 50% at 9 months. At baseline, 6 physical problems (pain, fatigue, sleep, dry/itchy skin, dry/obstructed nose and sexual problems), 2 emotional problems (worries, sadness), and 1 family problem (partner). At 9 months, only 2 problems remained in the physical section (indigestion, dryness, nasal obstruction).
Sica (2016) Case report [37]	Ileostomy. Woman, 28 years old. Symptoms began while she was studying to become an elementary school teacher. Prior to surgery she performed jazz, street, and contemporary dance. After stoma, she undertook running and classes at a local gym.	*Back to the sport*: When started to get her strength back, she began trying different classes at a local gym. Although she feels good, she tries to do the exercises in the least harmful way possible.
Isaacs (1984) Case series [40]	Ileostomy. Five men aged 22, 37, 40, 42, 56, and 56. Due to ulcerative colitis participated in the 1983 London Marathon. Questionnaires about ileostomy function, diet, training program, and experience. Urine and ileostomy secretion data were collected 3–4 h before the race and 5–6 h after. Before, they were weighed with running equipment, and blood pressure, heart rate, and a venous blood specimen were obtained. Procedure repeated within 5 min of the end of the race.	*Weight loss:* Between 1 and 3 kg. *Ileostomy flow*: The ileostomy discharged volume and sodium concentration showed no change during the race, but potassium concentration increased in contrast to the unchanged urine. Total water and potassium losses during the race were minimum. *Plasma biochemistry***:** Sodium levels lower. Urea and total protein higher compared to non-ostomized runners. After the race, increased blood urea, uric acid, and bilirubin occurred in all subjects.

**Table 4 sports-12-00273-t004:** Quantitative studies included in the review.

Author (Year) Design	Sample Size (*N*) Type of Stoma PA ^1^	Instruments	Statistical Data
Park et al. (2023) Cross-sectional [26]	*N* = 443 Male: *n* = 266 (61.9%); Female: *n* = 165 (37.9%); Non-binary *n* = 1 (0.2%). Urostomy: *n* = 212 (47.9%); Colostomy: *n* = 132 (36.1%); Ileostomy: *n* = 99 (22.3%) Parastomal hernia: No: *n* = 327 (75.3%); Yes: *n* = 129 (29.3%) PA: Not reported	**QoL** ^2^**:** *QoL scale*: Dimensions: 4; Reliability: Not reported; Likert scale: 4 items. **PA:** *International PA Questionnaire (IPAQ) Short Form:* Reliability: Not reported; Measure expressed in metabolic equivalents (MET minutes/week).	**QoL:** *QoL scale*: Total Lower if hernia (*U* = 11.99; *p* = 0.004). **PA:** *International PA Questionnaire Short Form:* PA: *U* = 8154; *Mean* = 579 (yes hernia) vs. 1689 (no hernia) *p* = 0.001. Correlation between PA intensity and time after ostomy making (*r* = 0.009; *p* = 0.870).
Goodman et al. (2022) Cross-sectional [27]	*N* = 1528 Male: *n* = 289 (20.4%); Female: *n* = 1122; (79.1%); Miss sex: *n* = 8 (0.6%) Ileostomy: *n* = 956 (67.7%); Colostomy: *n* = 444 (31.3%); Miss stoma: *n* = 19 (1.3%) PA: Not reported (number of days with exercise of 30 min or more on which the respiratory rate increased should be noted).	**QoL:** *Stoma Quality of Life (SQoL):* 19 items in 5 subscales (Work/social function; Sexuality and body image; Stoma function; Economic concerns; Skin irritation). Likert scale: 5 items. Score: 0 (never)–100 (always). Reliability (*α* = 0.89). **PA:** (additional item): number of days with increased respiratory frequency (Range: 0–7).	**QoL:** *SQoL [Mean (SD ^3^)]:* Work/social function: 63.6 (23.0); Sexuality and body image: 61.5 (19.3); Stoma function: 52.8 (20.6); Financial concerns: 81.3 (28.5); Skin irritation: 47.2 (27.9). **PA:** (days/week): *Mean* = 2.6 (*SD* = 2.3); 4 profiles according to SQoL responses: Profile 1: Good QoL (*n* = 891; 62.8%); Profile 2: Some problems with QoL (*n* = 184; 13.0%); Profile 3 Low QoL (*n* = 181; 12.8%); Profile 4: Financial concerns (n = 163; 11.5%). Individuals classified in Profile 3 were less able to stoma for more than 2 years (*OR*: 0.65; 95% *CI*: 0.43–0.96; *p* < 0.05) and spend more days physically active (*OR*: 0.85; 95% *CI*: 0.78, 0.94; *p* < 0.05), but were more likely to hernia (*OR*: 3.32; 95% *CI*: 2.17–5.07; *p* < 0.05).
Krogsgaard et al. (2022) Cross-sectional [28]	*N* = 571 Colorectal cancer and stoma in Denmark Colostomy: *n* = 491 (86%); Ileostomy: *n* = 80 (14%) PA: Not reported	**QoL:** *Colostomy Impact Score:* Ad hoc scale PROMIS items: 7 (odor, leakage, stool consistency, stoma site pain, skin problems, parasternal bulging, help with stoma management) Likert scale: not reported; Score: higher score more complications; Reliability: Not reported. **PA:** Compliant (Active, very active). Non-compliant (Inactive, insufficiently active).	**QoL:** *Colostomy Impact Score: n* = 313 (55%) higher impact of colostomy on PA; *n* = 358 (45%) lower impact of colostomy on PA. No association between colostomy impact and level of PA (*OR*: 1.59; 95% *CI*: 1.02–2.11). **PA:** Compliant (*n* = 293; 51%): Active *n* = 108 (36.86%); very active *n* = 185 (63.13%); >300 min *n* = 76 (41%); >151 min *n* = 59 (55%); >120 min *n* = 69 (37%); 60–120 min *n* = 36 (33%). Non-compliant (*n* = 278; 49%): Inactive *n* = 44 (15.8%); insufficiently active *n* = 234 (84.17%); <30–150 min *n* = 170 (73%); <59 min *n* = 55 (24%).
Mo et al. (2021) Cross-sectional [29]	*N* = 200 Male: *n* = 108; 54%; Female: *n* = 92; 46% Colostomy: *n* = 87; 43.5%; Ileostomy: *n* = 46; 23%; Urostomy: *n* = 67; 33.5% PA: Exercise Time recommendation according to intensity: 150 min per week if moderate. 75 min per week if vigorous.	**QoL:** *City of Hope Quality of Life Ostomy (COH-QoL-O*): Ad hoc questionnaire. Items (no scale) on diet: 8 general diet (high protein, low carbohydrate, fast food, diabetic, vegetarian, vegan, vegan, heart-healthy, no special diet); 5 broad diet (fast food, vegetarian/vegan, therapeutic/health-promoting, gastrointestinal symptom modulating, and no special diet) and a scale with 43 items Health-Related QoL with Likert scale not reported Score: 0 (worst QoL)–10 (best QoL). Reliability: Not reported. **PA:** *Self-Efficacy (SE) to Perfom Self-Management Behaviors*: Ad hoc questionnaire from the instrument (Godin Leisure-Time Exercise Questionnaire), contains 3 unscaled parts. 2 Items: “How confident do you feel you can perform gentle exercises for muscle strength and flexibility three to four times per week (range of motion, use of weights, etc.)?” and “How confident do you feel you can perform aerobic exercises such as walking, swimming or cycling three to four times per week?”. Likert scale: not reported. Score: 1 (not at all confident)–10 (totally confident); Reliability: (*α* = 0.83). Ad hoc *questionnaire:* One question with three parts on different PA intensities. Frequency and intensity of PA for one week. Measures: light (minimal exertion, no sweating), moderate (not strenuous, light sweating) and vigorous (heart beating fast, intense sweating).	**QoL:** *COH-QoL-O:* Those who met or exceeded American Cancer Society (ACS) guidelines reported greater psychological well-being (*Mean difference* = 1.03; 95% CI: 0.19–1.9); overall QoL (*Mean difference* = 0.74; 95% *CI*: 0.04–1.4) and physical strength (*Mean difference* = 1.29; 95% *CI*: 0.17- 2.4) compared to the non-active category. The group that met but not exceeded the ACS PA guidelines had better psychological well-being and physical strength scores that exceeded the minimally important difference compared to the non-active group. Total QoL score: (Non-active: *Mean* = 6.32 (*SD* = 0.18); Low active: *Mean* = 6.71 (*SD* = 0.26); Active: *Mean* = 7.08 (*SD* = 0.26)). Physical strength: (Non-active: *Mean* = 6.15 (*SD* = 0.30); Low active: *Mean* = 6.89 (*SD* = 0.40); Active: *Mean* = 7.48 (*SD* = 0.48). **PA:** *SE:* Self-efficacy for aerobic exercise accounted for greater variance for PA time (4.9%) than ostomy type (2.5%). Patients with urostomies who met ACS PA guidelines had higher self-efficacy scores for aerobic exercise capacity (*p* = 0.02). Patients who met ACS PA guidelines had higher self-efficacy scores for both gentle (8.1 points out of 10) and aerobic exercise (8.7 points out of 10) compared to those who did not meet guidelines. However, the latter group of patients reported moderate self-efficacy with respect to the ability to perform gentle (6.7 points) or aerobic (6.1 points) PA. Ad hoc *questionnaire:* No differences in PA intensity and type of ostomy: Intense PA (*p* = 0.06): Colostomy: *Mean* = 91.3 (*SD* = 198.2); Ileostomy: *Mean* = 38.6 (*SD* = 79.6); Vigorous PA (*p* = 0.08): Colostomy: *Mean* = 16.2 (*SD* = 63.3); Ileostomy: *Mean* = 3.8 (*SD* = 17.1); Moderate AF (*p* = 0.32): Colostomy: *Mean* = 59 (*SD* = 122.3); Ileostomy: *Mean* = 31 (*SD* = 66.6).
Lowe, Alsaleh, and Blake (2019) Cross-sectional [31]	*N* = 116 (completed questionnaire *n* = 94) Male: *n* = 46; 49%; Female: *n* = 48; 51% Colostomy, ileostomy and urostomy PA: Walking	**QoL:** *Patient Health Questionnaire (PHQ-9):* Items: 9 criteria for depression (depressed mood or irritability, decreased interest or pleasure, significant weight loss or loss of appetite, changes in sleep pattern, changes in activity, fatigue or loss of energy, guilt or worthlessness, concentration, and suicidality; Likert scale: 0 (never)–3 (almost every day); Score: 5–27 (higher score indicates depression, less than 4 indicates no depression) Reliability: Not reported. *SQoL:* Items: 20 in 5 domains (stoma device; sleep; sexual activity; relationships with family and friends and social relationships). Likert scale: 1 (always)–4 (never); Score: 20–80 (higher score indicates optimal QoL). Reliability: Not reported. *Social Physique Anxiety Scale (SPAS):* Items: 12; Likert scale: 1 (not at all characteristic of me)–5 (extremely characteristic of me). Score: Maximum 60 (low scores indicate reduced physical anxiety; Reliability: Not reported. **PA:** *IPAQ Short Form:* Short version. Measures of the amount of exercise in frequency (days per week) and duration (hours and minutes per day). Scale: Not reported; Score: Not reported; Reliability: Not reported. *Self-efficacy for exercise (SEE):* Items: 9 (environmental climate, boredom, pain, exercising alone, pleasant or unpleasant exercise, being busy, tiredness, stress, and depression). Scale: 0 (no confidence) to 10 (high confidence); Score: 0–90 (Higher score indicates more self-efficacy for exercise); Reliability: Not reported. *Excess Benefits and Barriers Scale (EBBS):* 29 items to measure perceived benefits and barriers to PA (Benefits: improved life, physical performance, psychological outlook, social interaction, and preventive health. Barriers: exercise environment, time investment; physical effort; and family discouragement). Likert scale: Benefits score 1 (strongly disagree) to 4 (strongly agree); for barriers inverse score. Score: 43–172, higher values indicate greater perception of PA benefits; Reliability: Not reported.	**QoL:** *PHQ-9:* Total PHQ-9: *Mean* = 3.22 (*SD* = 4.8). No differences for depression according to exercise intensity (*F* = 3.05; *p* = 0.53). *SQoL:* Total: *Mean* = 19.3 (*SD* = 12.9). No differences for exercise intensity (*F* = 0.40; *p* = 0.67). *SPAS:* No differences for physical anxiety according to exercise intensity (*F* = 1.97; *p* = 0.15). **PA:** *IPAQ:* Inactive (*n* = 36; 42%), minimally active (*n* = 35; 41%), active (*n* = 15; 17%) participants. Differences favor to women (more active) than men (*p* = 0.05). *SEE:* Total score (*Mean* = 40.8; *SD* = 20.7) indicated moderate self-efficacy for exercise. Scores had significant effect for MET intensity (*F* = 3.04, *p* < 0.001). Mean score for inactive group (*n* = 34, *Mean* = 30.4) was significantly lower than for the minimally active group (*n* = 33; *Mean* = 46; *p* = 0.03) and the active group (*n* = 17; *Mean* = 49; *p* = 0.01). There was no difference between the minimally active and active groups (*p* = 0.61). *EBBS:* The greatest barriers to PA were physical effort, time, and accessibility. Scores were significantly lower in inactive participants compared to minimally active and active (not reported *p*-value) individuals. Individuals with higher total score (higher perceived benefits) were more likely to be active or minimally active than inactive. Individuals with a lower score (fewer perceived benefits) were more likely to be inactive than minimally active or active. No differences between the minimally active group and the active groups (*p* = 0.90).
Kindred et al. (2019) RCT [32]	*N* = 46 IG: *N* = 20 Male: *n* = 8; 40%; Female: *n* = 12; 60% CG: *N* = 26 Male: *n* = 12; 46.2%; Female: *n* = 14; 53.8% PA: Strength exercise; moderate intensity (12 weeks). *Submaximal fitness test:* Treadmill walking. Participants select the fastest steady pace walking one mile by estimating maximal oxygen consumption. *Accelerometer data:* Moderate to vigorous PA was measured with the Computer Science and Applications, Inc. monitor 3 consecutive days.	**PA:** *Body Esteem Scale (BES):* 35 Items into subscales according to the sex of the individual (Men: physical attractiveness, upper body strength and physical condition. Women: sexual attractiveness, weight concern and physical condition). 5-point Likert scale: 1 (very negative feelings) 5 (very positive feelings). Score: Higher indicates positive body esteem. Reliability: Not reported. *Body composition:* Measured by electrical impedance with a single frequency current (50 kHz) produced by a Quantum II RJL analyzer (RJL Systems, Clinton Township, MI).	*BES:* Significant associations on physical fitness (body esteem) in men (3 months: *b* = 0.68; *SD* = 0.35; *p* = 0.04; 6 months: *b* = 1.36; *SD* = 0.66; *p* = 0.04; and 12 months: *b* = 0.84; *SD* = 0.36; *p* = 0.03). No statistical difference in women. *Body composition:* Among women there were positive associations between reductions in body fat and body size (3 months: *b* = 3.71; *SD* = 1.79; *p* = 0.05; 12 months: *b* = 5.99; *SD* = 2.95; *p* = 0.05).
Russell (2017) Cross-sectional [33]	*N* = 2631 Male: 46%; Female: 54%. Type of stoma: Not Specified PA: Not reported	Instrument not reported. Ad hoc survey with open and closed questions using a 3 and 5-point Likert scale.	People declared QoL “a little worse” than before surgery (22.5% diagnosis/suspected hernia vs. 16% without hernia; *p* < 0.05). 32% with hernia declare being “much less active” than before surgery compared to 19% without hernia (*p* < 0.001).
Russell (2017) Cross-sectional [34]	*N* = 2631 Male: 46%; Female: 54%. Type of stoma: Not Specified PA: Not reported	Ad hoc *survey*: “Living with a stoma, your experience” with open and closed questions, rating scales of 3 and 5-points Likert scale.	People with stoma reported being less active (24.8%) compared to other conditions (*p* < 0.001). People who reported performing PA since intervention (38%) showed a higher perceived QoL (*p* < 0.05).
Krouse et al. (2017) Cross-sectional [35]	*N* = 1063 (target population) rectal cancer survivors (<5 years after diagnosis) during 2010–2011. *N* = 557 (sample) Type of stoma: Not specified PA: Not reported.	**QoL:** *City of Hope QoL Ostomy (COH-QoL-O) (*Ad hoc *questionnaire)*: Items: Not specified. Scale: Not specified. Score: 0–10; higher score indicates better QoL. Subscales (physical, psychological, social, and spiritual well-being); Reliability: Not reported.	**QOL:** *COH-QoL-O:* Total QoL better in the group with guidelines with respect to not active (*Mean difference* = 0.43; 95% *CI*: 0.10–0.76). The group that followed the guidelines showed greater psychological well-being (*Mean difference* = 0.55; 95% *CI*: 0.23–0.88). Association between increased PA time and physical component (*Mean difference* = 6.0; 95% *CI*: 3.9–8.1), physical function (*Mean difference* = 7.0; 95% *CI*: 4.8–9.3), physical role (*Mean difference* = 4.5; 95% *CI*: 2.5–6.5), general health (*Mean difference* = 5.8; 95% *CI*: 3.5–8.2), vitality (*Mean difference* = 5.7; 95% *CI*: 3.6–7.8), social role (*Mean difference* = 3.7; 95% *CI*: 1.4–5.9) and emotional role (*Mean difference* = 3.8: 95% *CI*: 0.82–6.7). **PA:** *GLTEQ:* 34% (*n* = 190) not active, 26% (*n* = 145) insufficiently active, 13% (*n* = 72) meeting guidelines and 27% (*n* = 150) above guidelines. Relationship between PA time and being younger (*p* < 0.001), years since surgery (*p* = 0.02), college degree (*p* < 0.001), higher income (*p* = 0.004), married or partnered (*p* = 0.003), and comorbidity less than 2 (*p* < 0.001). No differences (*p* = 0.08) between having an ostomy and time of PA.
Courneya et al. (1999) Cohort [39]	*N* = 53 Males: 60%. Mean age: 60.7 years Type of stoma: Colostomy, ileostomy and urostomy. PA: Not reported.	**QoL:** *Functional Assessment of Cancer Therapy- Colorectal (FACT-C):* 6 dimensions: physical, functional, emotional, social, relationship with physician, additional aspects; Scale: Likert 0 (worst quality)–4 (best quality); Score: Not reported; Reliability: *α* = 0.85 (baseline); *α* = 0.91 (4 months). *Satisfaction with Life scale (SWL):* Items: 5 on unspecified aspects of life; Scale: Likert 1 (worst satisfaction)–7 (best satisfaction); Score: Not reported; Reliability: *α* = 0.91 (baseline); *α* = 0.92 (4 months). **PA:** *Godin Leisure Time Exercise Questionnaire (GLTEQ):* Items: 3 questions on frequency of days per week performing light, moderate, and strenuous exercise for 15 min. Scale: Open-ended.	**QoL:** *FACT-C:* Total: Baseline (*Mean* = 3.31; *SD* = 0.40). 4 months (*Mean* = 3.24; *SD* = 0.52). *Mean difference* = −0.07 (*SD* = 0.35). Physical: Baseline (*Mean* = 3.41; *SD* = 2.48). 4 months (*Mean* = 3.23; *SD* = 0.81). *Mean difference* = −0.18 (*SD* = 0.66). Functional: Baseline (*Mean* = 3.05; *SD* = 0.66). 4 months (*Mean* = 3.10; *SD* = 0.70). *Mean difference* = 0.05 (*SD* = 0.63). Emotional: Baseline (*Mean* = 3.37; *SD* = 0.60). 4 months (*Mean* = 3.49; *SD* = 0.50). *Mean difference* = −0.07 (*SD* = 0.35). Social: Baseline (*Mean* = 3.56; *SD* = 0.57). 4 months (*Mean* = 3.28; *SD* = 0.76). *Mean difference* = −0.28 (*SD* = 0.74). Changes in QoL at 4 months for social dimension (*p* < 0.008). *SWL:* Baseline (*Mean* = 4.98; *SD* = 1.51). 4 months (*Mean* = 5.00; *SD* = 1.41). *Mean difference* = 0.02 (*SD* = 1.12). Baseline correlation between satisfaction and functional dimension (*r* = 0.70). Multifactorial analysis indicated that functional dimension is the only one that explains 50% of variance: *F* (1.51) = 50.20; *p* < 0.001. Correlation at 4 months between satisfaction and functional (*r* = 0.73), emotional (*r* = 0.56), physical (*r* = 0.54), additional aspects (*r* = 0.49) and social (*r* = 0.40) dimensions. Multifactorial analysis: functional (*B* = 0.58), additional aspects (*B* = 0.25) and social (*B* = 0.19) dimensions explained 62% of variance: *F* (3.49) = 26.64; *p* < 0.001. Changes in satisfaction Baseline–4 months for physical (*r* = 0.38), functional (*r* = 0.33), and additional aspects (*r* = 0.31) explained 15% of variance: *F* (1.15) = 8.71; *p* < 0.005. **PA:** *GLTEQ:* Mild: Baseline (*Mean* = 2.64; *SD* = 2.48). 4 months (*Mean* = 2.75; *SD* = 2.52). *Mean difference* = 0.11 (*SD* = 2.77). Moderate: Baseline (*Mean* = 1.53; *SD* = 1.90). 4 months (*Mean* = 1.78; *SD* = 2.13). *Mean difference* = 0.25 (*SD* = 2.51). Extenuating: Baseline (*Mean* = 0.53; *SD* = 1.44). 4 months (*Mean* = 0.17; *SD* = 0.49). *Mean difference* = −0.36 (*SD* = 1.26). Increases in mild to moderate PA associated in total QoL at 4 months (*r* = 0.39; *p* < 0.01).

^1^ PA: Physical Activity; ^2^ QoL: Quality of Life; ^3^ SD: Standard Deviation.

## Data Availability

No new data were created.

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
