# Peer review of "Return to Physical Activity in Individuals with Surgical Stomas: A Scoping Review"

_sports, 2024, doi:10.3390/sports12100273_

Round 1
Reviewer 1 Report
Comments and Suggestions for Authors
This is a scoping review about Return to Sports Activity in Individuals with Surgical Stomas I find this an important topic, I find scoping reviews about quality of life challenging to write about but it is important because it forces surgeons to look beyond operating room and consequently visit their techniques and improve it.
the tables initially look long and difficult to follow but due to limited number of studies and importance of study and careful reading I finding them good summary of literature with clear points, the discussion is concise and to the point though when you mentioned biochemical profile of the patients with stoma I wanted to learn more how PA affects biochemical profile and how to take care of it .
the introduction is a long if you can make bit more concise
Author Response
We thank the reviewers and editors of Sports journal for their comments and suggestions for improvements. We have responded to each of them, which have clearly improved this new version of the manuscript. Modifications have been highlighted in red font in the text.
Reviewer 1
This is a scoping review about Return to Sports Activity in Individuals with Surgical Stomas I find this an important topic, I find scoping reviews about quality of life challenging to write about but it is important because it forces surgeons to look beyond operating room and consequently visit their techniques and improve it.
We appreciate the reviewer's comments and agree on the importance of making visible the results of surgical interventions on people's quality of life.
the tables initially look long and difficult to follow but due to limited number of studies and importance of study and careful reading I finding them good summary of literature with clear points, the discussion is concise and to the point though when you mentioned biochemical profile of the patients with stoma I wanted to learn more how PA affects biochemical profile and how to take care of it .
We agree with the reviewer regarding the length of the tables, but we consider it appropriate to extract all these results given that this is an exploratory review of a clinical aspect that has been little studied in the available literature. With regard to physical activity and biochemical profile, we have added some content and references in the discussion.
the introduction is a long if you can make bit more concise
Thank you for the suggestion. A few sentences have been cut (and revised the English) without altering the content of the introduction. It is not possible to make further adjustments without altering the sense of the introduction in order to provide a better context for potential readers of the manuscript.
Reviewer 2 Report
Comments and Suggestions for Authors
General comments
Thank you for providing me with the opportunity to review your Scoping Review. Overall, it is well-written and engaging. However, there are several points that need to be addressed:
Specific Comments
L26. Three out of your four keywords are included in the title. Please provide an alternative set of keywords using MeSH terms to enhance the visibility of your review.
L73-74. “In individuals with stomas, the return to sports activities is an indicator of QoL that reflects their well-being.” please include a reference to support this sentence.
L74-75. “Studies that have analyzed interventions promoting self-care in individuals with stomas suggest that it is feasible to increase their QoL, self-efficacy, and improve psychosocial health outcomes.” please, include a reference.
L87. Have there been other reviews conducted on this topic? Please specify this clearly in the introduction. If such reviews exist, provide a justification for the necessity of undertaking a scoping review.
L102. When was the registration completed? Please specify the date and indicate the phase of the review at that time.
L127. Please clearly identify the inclusion and exclusion criteria.
L128. Please provide additional information regarding the exclusion criteria.
L133. Please include additional information regarding the quality assessment in a separate section and provide relevant references to support this information.
L136. "As a scoping review, the critical appraisal process was not used to eliminate low-quality studies, but to identify and establish the quality of the included studies." please include a reference to support this sentence.
L138. Please provide more details regarding the "pilot phase"
L140. Please provide more information regarding the data extraction process. What methods were employed? Which bibliometric and sociodemographic variables were extracted?
L145. Please provide more information regarding “pilot phase”.
L147. The results section is quite superficial; please provide a more in-depth information of the results. This is a important issue.
L156. The authors do not provide information on the characteristics of the participants in the results section
L160. Please include the item-by-item quality assessment from JBI as supplementary material.
L179. In general, the discussion primarily summarizes the results of the included studies. A more thorough and in-depth analysis is required.
L268. What are the practical implications? What are the gaps in knowledge? Although the authors mention the need for further studies, it might be helpful to create a subsection that clearly identifies these issues for the reader.
Author Response
General comments
Thank you for providing me with the opportunity to review your Scoping Review. Overall, it is well-written and engaging.
We appreciate the reviewer's comments.
However, there are several points that need to be addressed:
Specific Comments
L26. Three out of your four keywords are included in the title. Please provide an alternative set of keywords using MeSH terms to enhance the visibility of your review.
We have used the descriptor terms as keywords; however, it is possible to add additional keywords; therefore, we have included two new MeSH: “Quality of Life” and “Ostomy”.
L73-74. “In individuals with stomas, the return to sports activities is an indicator of QoL that reflects their well-being.” please include a reference to support this sentence. L74-75. “Studies that have analyzed interventions promoting self-care in individuals with stomas suggest that it is feasible to increase their QoL, self-efficacy, and improve psychosocial health outcomes.” please, include a reference.
In response to these two above comments, we have merged the wording of these two sentences to reduce the length of the introduction and implemented two new references.
L87. Have there been other reviews conducted on this topic? Please specify this clearly in the introduction. If such reviews exist, provide a justification for the necessity of undertaking a scoping review.
No review on this issue has been published in the scientific literature, so the scoping review design is pertinent. The following sentence has been added to the introduction (lines 90-91): “Currently no reviews on PA and individuals with stomas have been developed”.
L102. When was the registration completed? Please specify the date and indicate the phase of the review at that time.
The date of registration has been added, indicating that it was before the start of the revision in the sub-heading design.
L127. Please clearly identify the inclusion and exclusion criteria.
Done.
L128. Please provide additional information regarding the exclusion criteria.
Done.
L133. Please include additional information regarding the quality assessment in a separate section and provide relevant references to support this information. L136. "As a scoping review, the critical appraisal process was not used to eliminate low-quality studies, but to identify and establish the quality of the included studies." please include a reference to support this sentence.
In relation to the two above comments, reference 24 has been added to justify that scoping reviews do not require a formal analysis of the quality of evidence. We have performed this analysis of the evidence for the purpose of simply reporting information on the quality of the studies, without establishing criteria for excluding studies on quality grounds. Therefore, we consider that establishing a specific sub-heading to assess the quality of the evidence is more appropriate for systematic review designs, not for scoping reviews.
L138. Please provide more details regarding the "pilot phase". L145. Please provide more information regarding “pilot phase”.
We appreciate the reviewer's interest in proposing improvements to the study methodology. The piloting consists of conducting a first assessment with one of the JBI tools for each included study design (in the case of critical appraisal) and data extraction in a first study for each included study design (in the case of data extraction). We believe that the sentence included in the methodology is sufficient for readers to understand the methodological process followed. In our opinion, further information is irrelevant.
L140. Please provide more information regarding the data extraction process. What methods were employed? Which bibliometric and sociodemographic variables were extracted?
Done.
L147. The results section is quite superficial; please provide a more in-depth information of the results. This is a important issue.
Thank you very much for this suggestion. Done.
L156. The authors do not provide information on the characteristics of the participants in the results section
After responding to the previous comment, it is possible that the content of this comment has been improved. However, characteristics of the included studies (author, year, design, country, aim, conclusions) have been described as this is a review study, and the following tables detail patient´s characteristics.
L160. Please include the item-by-item quality assessment from JBI as supplementary material.
Attached as a supplementary file (supplementary table 2).
L179. In general, the discussion primarily summarizes the results of the included studies. A more thorough and in-depth analysis is required.
We have implemented more in-depth discussion and added new references.
L268. What are the practical implications? What are the gaps in knowledge? Although the authors mention the need for further studies, it might be helpful to create a subsection that clearly identifies these issues for the reader.
At the end of the discussion we have implemented new limitations and additional strengths. In the conclusions we have implemented implications for clinical practice.
Reviewer 3 Report
Comments and Suggestions for Authors
Thank you for the opportunity to review this paper. Aim of synthesizing the available evidence regarding the return to physical activity and sports practice in individuals with surgical stomas, a scoping review was developed following the methodological approach of the Joanna Briggs Institute and the Preferred Reporting Items for Systematic reviews and Meta-Analyses for Scoping Reviews criteria. I congratulate the authors on a work that has been done so well. While reviewing the manuscript, I had only two comments, which I have included below.
· Introduction: „The available literature on returning to sports activities after surgery is extensive, particularly regarding musculoskeletal and surgical issues in the fields of pediatric and sports traumatology.” - please provide some literature references
· Line 208: Research articles usually do not use the word „we”.
Author Response
We thank the reviewers and editors of Sports journal for their comments and suggestions for improvements. We have responded to each of them, which have clearly improved this new version of the manuscript. Modifications have been highlighted in red font in the text.
Thank you for the opportunity to review this paper. Aim of synthesizing the available evidence regarding the return to physical activity and sports practice in individuals with surgical stomas, a scoping review was developed following the methodological approach of the Joanna Briggs Institute and the Preferred Reporting Items for Systematic reviews and Meta-Analyses for Scoping Reviews criteria. I congratulate the authors on a work that has been done so well. While reviewing the manuscript, I had only two comments, which I have included below.
We thank reviewer 3 for his comments on the manuscript and for considering it an interesting review.
Introduction: „The available literature on returning to sports activities after surgery is extensive, particularly regarding musculoskeletal and surgical issues in the fields of pediatric and sports traumatology.” - please provide some literature references
We appreciate this valuable advice; we have included some appropriate and up-to-date references to support these affirmations.
Line 208: Research articles usually do not use the word „we”.
We appreciate this comment. It was a mistake to include this pronoun in the text. We have adjusted the sentence with more scientific and impersonal language.
Reviewer 4 Report
Comments and Suggestions for Authors
I read with interest the manuscript entitled "Return to Sports Activity in Individuals with Surgical Stomas: A Scoping Review".
The MeSH terms within the abstract do not match those listed in the manuscript. I suggest that you do not list all the MeSH terms in the abstract, nor the designs of the included studies. Please give more specific conclusions in the abstract.
I suggest that you use MeSH terms for keywords, preferably more than the above four.
I suggest that you write the introduction more concisely and shorten it. Also, at the beginning you focused on cancer as the cause of surgical procedures that lead to the formation of a stoma. Please emphasize and touch on a number of conditions, from the earliest age, that can result in surgical intervention in the form of stoma formation.
The question is clumsily formulated, that is, it is formulated too general and broad.
Why haven't you registered your scoping review in PROSPERO? I suggest you register and ask for a registration number.
The MeSH terms listed in the text differ from those in the tables. You must clearly state that you have used all derivatives of the terms.
Submit table 1 as suppl. material and not as part of the manuscript.
Give summary criteria for the participants or populations being studied by the review.
You emphasize "colostomy" and "ileostomy". What other stomas are there? Why did you decide only on colostomies and ileostomies?
Give the pre-specified primary (most important) and secondary (additional) outcomes of the review, including details of how the outcome is defined and measured and when these measurements are made.
Did study selection and/or data extraction was blinded (researchers unaware of author/journal details) and whether and how authors of eligible studies were contacted to provide missing or additional data?
How did you assess the risk of bias? Please explain in detail and include the above in the manuscript.
In Figure 1, the data is inconsistent (eg 826+3+15 is 844, not 843). Please check all numbers in detail.
Also, in the frame of Figure 1, you incorrectly exclude data, for example, you have gray literature in two places. Please check in detail how the flowchart is filled out! There are guidelines. study them and make a new flowchart.
I suggest that you provide the full references of the articles in Supplementary Table S1.
I suggest that you present Tables 3 and 4 in text, as part of the manuscript results, and submit Table 5 as a suppl. material.
I ask that you start the discussion by presenting the answer to the question posed at the end of the introduction, as well as the most relevant results, which you then elaborate on in detail.
Most of the discussion is about potential limitations, and little is done about the insights you should have gleaned from the collected articles. I suggest you revise the discussion thoroughly.
Please recognize all the strengths and limitations of your study and include them in the manuscript.
The conclusion is written universally and generally. Please be concise with a clear message after the scoping review. What message are you sending readers?
References are not written in accordance with the instructions for authors. Correct them.
Comments on the Quality of English LanguageModerate editing of English language required.
Author Response
I read with interest the manuscript entitled "Return to Sports Activity in Individuals with Surgical Stomas: A Scoping Review".
The authors appreciate this comment from reviewer 4. Thank you very much.
The MeSH terms within the abstract do not match those listed in the manuscript. I suggest that you do not list all the MeSH terms in the abstract, nor the designs of the included studies. Please give more specific conclusions in the abstract.
Done. We are particularly grateful for the suggestion to improve the content of the conclusions.
I suggest that you use MeSH terms for keywords, preferably more than the above four.
Done.
I suggest that you write the introduction more concisely and shorten it. Also, at the beginning you focused on cancer as the cause of surgical procedures that lead to the formation of a stoma. Please emphasize and touch on a number of conditions, from the earliest age, that can result in surgical intervention in the form of stoma formation.
This revision has already been made. The required changes have been commented on in several suggestions from other reviewers above.
The question is clumsily formulated, that is, it is formulated too general and broad.
We appreciate this comment by the reviewer. As this is a scoping review question, we have implemented the Population, Concept, and Context (PCC) structure, which we have added in the materials and methods (design sub-heading) as follows: Population (P): Individuals with surgical stomas; Concept (C): Return to sports and physical exercise; and Context (C): Post-surgery rehabilitation.
Regarding the review question, the JBI manual points out that the outcomes, interventions or phenomena of interest need not be explicit for a scoping review; however, elements of each of these may be implicit in the concept under review at:
https://jbi-global-wiki.refined.site/space/MANUAL/355862667/10.2.2+Developing+the+title+and+question
Why haven't you registered your scoping review in PROSPERO? I suggest you register and ask for a registration number.
We appreciate this comment, but PROSPERO does not register scoping reviews; we have therefore registered the protocol in OSF, as indicated in the design sub-heading.
The MeSH terms listed in the text differ from those in the tables. You must clearly state that you have used all derivatives of the terms.
We apologise for this error in the drafting of the report and appreciate this important contribution to clarifying the methodology. The searches are correct with the terms included in the strategies in table 1; therefore, we have added in the text all MeSH used in the searches and specified that other entry terms or synonyms have been used.
Submit table 1 as suppl. material and not as part of the manuscript.
Following the reviewer's suggestion, table 1 has been removed as supplementary file.
Give summary criteria for the participants or populations being studied by the review.
Done. We have fine-tuned and further specified the inclusion and exclusion criteria.
You emphasize "colostomy" and "ileostomy". What other stomas are there? Why did you decide only on colostomies and ileostomies?
In the introduction it has been specified that colostomies and ileostomies are the two main types of stomas, included urinary tract. In addition, MeSH terms referring to other types of stoma have been included in the methodology to indicate that they have been searched for as additional terms.
Give the pre-specified primary (most important) and secondary (additional) outcomes of the review, including details of how the outcome is defined and measured and when these measurements are made.
We have specified this information in the methodology (data extraction subheading).
Did study selection and/or data extraction was blinded (researchers unaware of author/journal details) and whether and how authors of eligible studies were contacted to provide missing or additional data?
We have specified this information in the methodology (data extraction subheading). However, it has not been necessary to contact the authors.
How did you assess the risk of bias? Please explain in detail and include the above in the manuscript.
No risk of bias assessment has been carried out. Only critical assessment has been performed with JBI tools. This comment has also been responded to the reviewer 2 with the following text: reference 24 has been added to justify that scoping reviews do not require a formal analysis of the quality of evidence. We have performed this analysis of the evidence for the purpose of simply reporting information on the quality of the studies, without establishing criteria for excluding studies on quality grounds. Therefore, we consider that establishing a specific sub-heading to assess the quality of the evidence is more appropriate for systematic review designs, not for scoping reviews.
In Figure 1, the data is inconsistent (eg 826+3+15 is 844, not 843). Please check all numbers in detail.
We apologise for ambiguously explaining this aspect in the diagram, but the figures are correct as 843 corresponds to excluded due to inclusion criteria and 1 (in the top line of the diagram) corresponds to excluded due to methodology (843+1=844). To correct possible confusion, we have deleted the number 843 and added the number 844 at the beginning of the records excluded.
Also, in the frame of Figure 1, you incorrectly exclude data, for example, you have gray literature in two places. Please check in detail how the flowchart is filled out! There are guidelines. study them and make a new flowchart.
The flow chart is correct, grey literature records have been excluded in both screening phases. First 15 (grey literature) were removed in the screening by title and abstract; subsequently 3 grey literature records were removed in the full-text screening. The reason for this is that in the screening by title and abstract it was not detected that they were grey literature and it was preferred to keep them for subsequent full-text screening, to be eliminated in this step. We kindly ask the reviewer to provide us a reference with the guidelines for this process. Nevertheless, we have changed Figure 1 to editable format for future modifications that may need to be made.
I suggest that you provide the full references of the articles in Supplementary Table S1.
If the reviewer looks at the content of the table, he/she includes all the essential elements in a reference. This is why we have included the DOI. We appreciate the suggestion but consider that this modification in a supplementary table is not appropriate. In other reviews published in this journal or in MDPI by our team, it has never been necessary to include the complete reference.
I suggest that you present Tables 3 and 4 in text, as part of the manuscript results, and submit Table 5 as a suppl. material.
After removing table 1 (searches in databases) to the supplementary material, we have adjusted the numeration of the tables. With regard to moving table 5 (now table 4) as supplementary material, we consider that this is not appropriate as these are results of the review and need to be available in the text of the manuscript.
I ask that you start the discussion by presenting the answer to the question posed at the end of the introduction, as well as the most relevant results, which you then elaborate on in detail.
Done. We thank reviewer 4 for this comment.
Most of the discussion is about potential limitations, and little is done about the insights you should have gleaned from the collected articles. I suggest you revise the discussion thoroughly.
We appreciate this comment from the reviewer regarding the limitations of the research. The discussion addresses important aspects of the available evidence, which is scarce and of limited quality. However, this does not imply limitations of the review process, which have been stated at the end of the discussion. In any case, we have implemented improvements in the discussion, including the incorporation of some strengths of the study. To enhance the discussion we have implemented content relating to key aspects of stomas and their complications in relation to quality of life.
Please recognize all the strengths and limitations of your study and include them in the manuscript.
Done.
The conclusion is written universally and generally. Please be concise with a clear message after the scoping review. What message are you sending readers?
We have redefined the conclusions according to the reviewer's recommendations.
References are not written in accordance with the instructions for authors. Correct them.
Corrected. Thank you very much for the suggestion to improve the citation style.
Moderate editing of English language required.
Improvements have been implemented in the English of the manuscript.
Round 2
Reviewer 2 Report
Comments and Suggestions for Authors
L98. To improve the readability of your work, I recommend that you include subsection titles, such as "2.1 Design," and place the text on a separate line.
L138. Please provide more details regarding the "pilot phase".
L145. Please provide more information regarding “pilot phase”.
We appreciate the reviewer's interest in proposing improvements to the study methodology. The piloting consists of conducting a first assessment with one of the JBI tools for each included study design (in the case of critical appraisal) and data extraction in a first study for each included study design (in the case of data extraction). We believe that the sentence included in the methodology is sufficient for readers to understand the methodological process followed. In our opinion, further information is irrelevant.
Comment 2: I disagree with the authors. The methodology section must be detailed with the utmost precision and clarity to allow for the replication of your work and to minimize publication bias as much as possible. Although pilot phases are common, most scoping reviews do not include them, and the reader may not be familiar with this. Please provide as much detail as possible regarding this phase.
Author Response
We thank the reviewers and editors of Sports journal for their comments and suggestions for improvements. We have responded to each of them, which have clearly improved this new version of the manuscript. Modifications have been highlighted in red font in the text.
Reviewer 2
L98. To improve the readability of your work, I recommend that you include subsection titles, such as "2.1 Design," and place the text on a separate line.
Done. Thank you very much for the suggestions in this regard.
L138. Please provide more details regarding the "pilot phase".
L145. Please provide more information regarding “pilot phase”.
We appreciate the reviewer's interest in proposing improvements to the study methodology. The piloting consists of conducting a first assessment with one of the JBI tools for each included study design (in the case of critical appraisal) and data extraction in a first study for each included study design (in the case of data extraction). We believe that the sentence included in the methodology is sufficient for readers to understand the methodological process followed. In our opinion, further information is irrelevant.
Comment 2: I disagree with the authors. The methodology section must be detailed with the utmost precision and clarity to allow for the replication of your work and to minimize publication bias as much as possible. Although pilot phases are common, most scoping reviews do not include them, and the reader may not be familiar with this. Please provide as much detail as possible regarding this phase.
Done. We have provided a detailed description of the piloting process for each of the points indicated.
Reviewer 4 Report
Comments and Suggestions for Authors
Thank you for the answers and clarifications.
Within the flowchart, how many were excluded in the end due to Methodology and Inclusion criteria? (881 - 844 or 845?). What about Methodology?
References are not written in accordance with the instructions for authors. Correct them.
Comments on the Quality of English LanguageModerate editing of English language required.
Author Response
Thank you for the answers and clarifications.
Within the flowchart, how many were excluded in the end due to Methodology and Inclusion criteria? (881 - 844 or 845?). What about Methodology?
The correct is 881 – 844= 37.
We would like to thank the reviewer once again for emphasizing this point in the flowchart and helping to clarify the issue. We have verified that the record eliminated due to methodology was due to an opinion article (not a research study) and that it is not grey literature either. Additionally, we had erroneously included the record twice under grey literature, which was incorrect. The number of records excluded due to grey literature has been corrected to n=14. We have updated the flowchart to reflect this new criterion (No research study) and removed the inappropriate criterion. Thank you for your valuable feedback.
References are not written in accordance with the instructions for authors. Correct them.
Revised.
English: Revised.